# Efficacy of Immune Checkpoint Blockade and Biomarkers of Response in Lymphoma: A Narrative Review

**DOI:** 10.3390/biomedicines11061720

**Published:** 2023-06-15

**Authors:** Sarah Perdikis-Prati, Semira Sheikh, Antonin Bouroumeau, Noémie Lang

**Affiliations:** 1Department of Oncology, Geneva University Hospital, 1205 Geneva, Switzerland; sarah.prati@gmail.com; 2Department of Hematology, Universitätsspital Basel, 4031 Basel, Switzerland; semirasheikh@googlemail.com; 3Division of Clinical Pathology, Diagnostic Department, Geneva University Hospital, 1206 Geneva, Switzerland; antonin.bouroumeau@hcuge.ch; 4Center of Translational Research in Oncohematology, Faculty of Medicine, University of Geneva, 1206 Geneva, Switzerland

**Keywords:** Hodgkin lymphoma, non-Hodgkin lymphoma, PD-1/PD-L1 checkpoint inhibitors, tumor mutational burden, predictive biomarkers

## Abstract

Immune checkpoint blockade (ICB) has revolutionized the prognosis of several advanced-stage solid tumors. However, its success has been far more limited in hematological malignancies and is mostly restricted to classical Hodgkin lymphoma (cHL) and primary mediastinal B cell lymphoma (PMBCL). In patients with non-Hodgkin lymphoma (NHL), response to PD-1/PD-L1 ICB monotherapy has been relatively limited, although some subtypes are more sensitive than others. Numerous predictive biomarkers have been investigated in solid malignancies, such as PD-L1 expression, tumor mutational burden (TMB) and microsatellite instability (MSI), among others. This review aims to appraise the current knowledge on PD-1/PD-L1 ICB efficacy in lymphoma when used either as monotherapy or combined with other agents, and describes potential biomarkers of response in this specific setting.

## 1. Introduction

The tumor microenvironment (TME), consisting of T cells, tumor-associated macrophages (TAMs), dendritic cells (DCs), neutrophils and natural killer (NK) and stromal cells, is thought to play a significant role in the development and progression of many cancers, as well as in tumor escape from the immune system. Accumulation of somatic mutations during oncogenesis has been shown to result in the presentation of neoantigens at the tumor cell surface, which can elicit tumor-specific CD4+ and CD8+ T cells with antitumor potential [1]. Many studies have demonstrated that tumor evasion is, at least in part, mediated by the inhibition of antitumor T-cell responses, mostly via upregulation of immune checkpoint molecules [2,3,4]. The understanding of these resistance mechanisms led to the delineation of the concept of immune checkpoint blockade (ICB).

More precisely, interaction between the immune checkpoint (IC) molecule programmed cell death protein 1 (PD-1) expressed by activated tumor-infiltrating T cells and its ligands (PD-L1 and PD-L2) expressed by the surrounding tumor and TME cells commonly leads to downregulation of neoantigen-specific T-cell responses. Blocking these interactions is a frequently used therapeutic approach to restore the antitumor effect of the host immune system [5,6].

In lymphoproliferative disorders, PD-1 is frequently expressed on tumor cells themselves as in tumor-infiltrating lymphocytes (TILs), while its ligands may be upregulated by tumor cells (some B-cell or T-cell lymphomas) but also TME cells such as TAMs, mast cells and mesenchymal cells [7,8,9]. PD-1 blockade using nivolumab or pembrolizumab has dramatically improved the prognosis of relapsed/refractory (R/R) classical Hodgkin lymphoma (cHL) and is now a well-recognized therapeutic option in this setting [10,11,12]. On the other hand, the efficacy of PD-1 or PD-L1 blockades in non-Hodgkin lymphoma (NHL) has shown disappointingly low response rates, except for some specific subsets of NHL, such as primary mediastinal B cell lymphoma (PMBCL) [13], primary testicular lymphoma (PTL) or primary central nervous system lymphoma (PCNSL) [14,15]. Interestingly, these NHL subsets have been shown to be highly infiltrated by T cells [16,17]. Of note, the different anti-PD-1/PD-L1 antibodies used in the clinic vary in their IgG isotypes and affinity to the various Fc gamma receptors (FcγRs) expressed on immune cells (Table 1); however, whether these changes translate into different clinical benefits is not well established [18]. A persistent challenge remains the identification of predictive biomarkers of response to PD-1/PD-L1 ICB in this setting.

Here, we present a review of the clinical efficacy of PD-1/PD-L1 ICB as monotherapy and in combination with other agents for different lymphoma subtypes.

## 2. Clinical Efficacy of ICB in Lymphomas

### 2.1. Classical Hodgkin Lymphoma (cHL)

#### 2.1.1. PD-1/PD-L1 ICB as Monotherapy

The safety and activity of PD-1 ICB was first tested by Ansell et al. (2015) in a phase 1 study on 23 heavily pretreated cHL patients who received biweekly nivolumab, a PD-1 inhibitor, demonstrating an impressive 87% overall response rate (ORR) and a 17% complete metabolic response rate (CMR) [11,29]. Larger studies subsequently confirmed the efficacy of PD-1 ICB monotherapy in R/R cHL using either nivolumab or pembrolizumab (Table 2) [21,30,31,32]. All trials showed similar results in R/R cHL patients, including those relapsing after autologous stem cell transplantation (ASCT) and those ineligible for ASCT (Table 2). At a 5-year follow-up, both CheckMate 205 and KEYNOTE-087 trials demonstrated sustained responses, with median PFSs of 15 months and 13.7 months, respectively. Additionally, a subgroup analysis of the KEYNOTE-087 trial demonstrated the efficacy of pembrolizumab after treatment with the anti-CD30 antibody drug conjugate brentuximab vedotin (BV) (Table 2) [33]. Based on these results, the FDA approved nivolumab (2016) and pembrolizumab (2017) for R/R cHL patients relapsing after ASCT and BV [34]. More recently, the phase 3 KEYNOTE-204 study (2021) compared pembrolizumab and BV in patients with R/R cHL either following ASCT or in patients ineligible for ASCT and showed an increased median progression-free survival (PFS) for pembrolizumab over BV (Table 2) [35].

In addition to nivolumab and pembrolizumab, several other PD-1 inhibitors have been investigated in R/R cHL early-phase trials. PD-1/PD-L1 antibodies vary in their IgG isotypes (Table 1). PD-1 IgG4 antibodies include nivolumab, pembrolizumab, sintilimab, camrelizumab, tislelizumab and zimberelimab, with ORRs ranging from 42% to 91% [24,25,26,27,40,41,42,43,44], while penpulimab, an IgG1 antibody, demonstrated an ORR of 89% with a 47% CR in a multicenter phase 1/2 trial [28]. To our knowledge, avelumab, an IgG1 anti-PD-L1 antibody, is the only anti-PD-L1 antibody that has been tested in this setting, achieving a 42% ORR and a 19% CR [39] (Table 2).

#### 2.1.2. PD-1/PD-L1 ICB in Combination with Chemotherapy

##### Frontline Setting

Assuming that PD-1/PD-L1 ICB may prime the TME for the induction of antitumor T-cell responses before patients receive cytotoxic agents, both sequential and concomitant combinations of ICB with conventional chemotherapy have been evaluated in the frontline setting. Ramchandren et al. (2019) evaluated a sequential approach of four cycles of single-agent nivolumab followed by twelve cycles of nivolumab given concomitantly with doxorubicin, vinblastine and dacarbazine (N-AVD) in patients with newly diagnosed advanced-stage cHL (cohort D of CheckMate 205) (Table 3) [61]. N-AVD has also been investigated in early-stage unfavorable cHL in a phase 2 trial conducted by the German Hodgkin Study Group (2020), achieving similar response rates (Table 3) [62]. N-AVD regimen is currently being evaluated in a frontline randomized phase 3 study against BV-AVD in patients with advanced-stage cHL, with an estimated completion date in 2024 (NCT03907488). Similarly, a sequential strategy of pembrolizumab monotherapy followed by combination with AVD has been evaluated in unfavorable or advanced-stage cHL patients (Table 3) [63]. Another ongoing nonrandomized PET-adapted phase 2 study (NCT03617666) is investigating the safety and efficacy of sequential avelumab followed by ABVD in first-line high-risk cHL [64]. As approximatively twenty percent of newly diagnosed cHL patients are ineligible for intensive chemotherapy regimens and consequently at risk of experiencing worse outcomes [65,66], Cheson and colleagues (2020) investigated the combination of 8 cycles of BV with nivolumab for older (>60 years) or chemo-ineligible newly diagnosed cHL patients. The trial was prematurely closed after an interim analysis failed to show that the combination met the predefined ORR criteria (ORR > 68%); however, 61% of all evaluable patients displayed an objective response and 48% achieved CMRs, demonstrating that this well-tolerated combination is active in this frail population (Table 3) [67].

##### Relapsed/Refractory Setting

Herrera and colleagues (2018) evaluated a nivolumab and BV combination for up to four cycles in the first-salvage-setting phase 1/2 study; this regimen was demonstrated to be well tolerated, achieving an ORR of 82%, including a 61% CMR [75]. An extended 3-year follow-up confirmed the durability of responses, with an estimated 77% 3-year PFS after a median follow-up of 34.3 months [76]. A phase 3 trial further investigating this regimen versus BV alone in R/R or ASCT-ineligible cHL patients was closed due to insufficient enrolment (NCT03138499). Three other trials demonstrated the efficacy of PD-1 ICB combined with standard salvage chemotherapy (ICE or gemcitabine, vinorelbine and liposomal doxorubicin (GVD) prior to ASCT) (Table 3) [68,69,70,77]. Atezolizumab, a PD-L1 inhibitor, scarcely tested as monotherapy in lymphoma ([59], NCT03120676), is currently being investigated in combination with a BeGEV regimen (bendamustine, gemcitabine and vinorelbine) (NCT05300282) (Table 3).

Patients with R/R HL who undergo ASCT have an expected 60% 18-month PFS [112,113,114]. Lepik et al. evaluated the utility of nivolumab and bendamustine administered for up to three cycles in heavily pretreated R/R cHL patients who had failed at least two lines of prior therapy, including nivolumab monotherapy. Among all enrolled patients (n = 30), 26 achieved a response (ORR: 87%) and 17 a CMR (57%) (Table 2) [74]. To our knowledge, only one trial phase 2 (Armand et al.) investigated the role of PD-1/PD-L1 ICB (pembrolizumab) as a consolidative therapy in the post-ASCT setting, achieving PFS and OS rates at 18 months of 82% and 100%, respectively (Table 2) [45].

#### 2.1.3. PD-1/PD-L1 ICB Combined with Other Agents

The combination of nivolumab with other ICBs, such as ipilumumab, an anticytotoxic T-lymphocyte-associated protein 4 (CTLA4) inhibitor, or lirilumab, an antibody targeting the killer cell Ig-like receptors (KIR) expressed by NK cells, did not seem to significantly improve the response rate in 31 R/R cHL patients enrolled in phase 1 of the CheckMate 039 trial (2016, 2021) (Table 3) [19,81]. The safety and efficacy of a triple combination of nivolumab, ipilimumab and BV was tested in a phase 1/2 trial conducted by Diefenbach et al. (2020) on patients with R/R cHL in comparison with nivolumab-BV or ipilimumab-BV. The triple regimen was associated with increased toxicity without clinical benefit (Table 3) [71]. Potentially more promising are PD-1/PD-L1 ICB and histone deacetylase inhibitor (HDACi) combinations. The combination of pembrolizumab with entinostat was investigated in 22 R/R cHL patients by Sermer et al. (2020, 2021) in a phase 2 trial with an ORR of 86% (Table 3) [72,73]. Similarly, a phase 2 trial with camrelizumab, another PD-1 inhibitor, combined with decitabine, demonstrated improved response rates (ORR 95%, CMR 79%) compared with camrelizumab monotherapy (ORR 89%, CMR 32%) (Table 3) [42,43].

Various combinations of PD-1/PD-L1 ICB with novel immunotherapies are currently under investigation in early-phase trials. A first report of pembrolizumab combined with AFM-13, a CD30-CD16 bispecific antibody stimulating innate immune cells, such as NK and macrophages, achieved an ORR of 83% in R/R cHL patients who had received a median of three prior lines of therapy [78,79]. A recent phase 1 study evaluating the benefit of adoptive cellular therapy consisting of tumor-associated antigen (TAA)-specific T cells enrolled 10 patients with R/R HL (n = 8 active disease, n = 2 adjuvant after ASCT) to receive TAA-specific T cells (autologous or allogenic) with nivolumab given as a priming agent in 6 out of the 10 patients. Among the patients with active disease, one patient achieved CMR and seven had stable disease (SD) at 3 months [78]. In addition, there is growing evidence that chimeric antigen receptor (CAR) T-cell fitness may be improved by the adjunction of PD-1 ICB. With the limitation of a small sample size (n = 12), PD-1 ICB administration after CD30 CAR T-cell therapy in CD30+ lymphoma patients (n = 9 cHL, n = 1 angioimmunoblastic T-cell lymphoma, n = 2 gray zone lymphoma) suggested improved efficacy (ORR 86% versus 100%; CR 27% versus 80%) (Table 3) [111]. Similarly, the role of anti-PD-1 therapy after CD30 CAR T-cell treatment is currently being evaluated in R/R cHL patients (NCT04134325) [115,116]. Recently, Timmerman et al. (2022) demonstrated that the association of favezelimab (lymphocyte activating gene-3 (LAG-3) ICB) and pembrolizumab could be an effective therapeutic option for patients progressing under PD-1 ICB (ORR 31%, CR 7%) [82]. Other combination approaches in the R/R cHL setting are ongoing, such as nivolumab with ruxolitinib (NCT03681561) and nivolumab/pembrolizumab combined with radiation (NCT04419441).

### 2.2. Non-Hodgkin Lymphoma (NHL)

#### 2.2.1. PD-1/PD-L1 ICB as Monotherapy

##### Aggressive NHL

First tested on a range of various hematological malignancies, nivolumab achieved an ORR of 36% and a CMR of 18% in the 11 R/R diffuse large B cell lymphoma (DLBCL) patients enrolled in a phase 1 study [53]. Single-agent activity was further evaluated in DLBCL patients who were relapsing after ASCT or ineligible for ASCT [49] and in patients with various hematologic malignancies relapsing after allogeneic stem cell transplantation (allo-SCT) [55], demonstrating modest benefits (Table 2). In contrast to cHL, maintenance treatment with pembrolizumab administered after ASCT in DLBCL and PMBCL patients did not show any clinical benefit [54]. A randomized phase 3 trial investigating tislelizumab, a PD-1 inhibitor, as maintenance in DLBCL after ASCT is planned (NCT04799314). Even though the benefit of PD-1/PD-L1 inhibitors as monotherapy in R/R aggressive NHL has been disappointing, better activity has been observed in patient subsets, such as PMBCL, PCNSL and PTL patients [14,15,53]. Benefits from pembrolizumab were demonstrated in R/R PMBCL with 45–48% ORR and 13–33% CMR, leading to the accelerated approval of pembrolizumab by the FDA in this setting (Table 2) [32,117]. Nivolumab demonstrated activity in four patients with R/R PCNSL and one patient with PTL CNS recurrence; all five patients had clinical and radiographic responses to the monotherapy [15]. The efficacy in these specific aggressive NHL subgroups is likely due to their particular biology. This is described in further detail in the biomarker section. Pembrolizumab has also been tested on nine patients experiencing Richter transformation (RT) (Table 2) [56].

PD-1/PD-L1 ICB demonstrated modest results in natural killer NK/T-cell NHL. In a phase 1 trial, nivolumab monotherapy achieved no observed CR (Table 2) [53]. On the other hand, pembrolizumab showed slightly improved activity in cutaneous T-cell lymphoma (CTCL) [57] and in a series of seven R/R extranodal NK/T-cell lymphoma (ENKTL) patients, with two of them achieving CR [118]. This initial signal of activity in R/R ENKTL was recently confirmed by two phase 2 trials evaluating avelumab, a PD-L1 inhibitor, and sintilimab, a PD-1 inhibitor [51,52] (Table 2).

##### Indolent B NHL

PD-1/PD-L1 ICB has also been evaluated in indolent lymphomas. Lesokhin et al. (2016) administered single-agent nivolumab to 10 R/R FL patients (ORR 40%) [53], but this result was not confirmed in a larger phase 2 study conducted by Armand et al. (CheckMate-140) (ORR 4%) (Table 2) [48]. To date, other indolent lymphoid malignancies such as chronic lymphocytic leukemia (CLL), marginal cell lymphoma (MZL) and Waldenström macroglobulinemia (WM) have not shown significant response rates with PD-1/PD-L1 ICB (Table 2) [56,105,110].

#### 2.2.2. PD-1/PD-L1 ICB in Combination with Other Agents

##### Aggressive NHL

Frontline setting

PD-1/PD-L1 ICB does not seem to add much benefit to frontline immunochemotherapy in newly diagnosed DLBCL. The combination of atezolizumab with frontline R-CHOP was tested on DLBCL patients [89,90] as a combination of pembrolizumab with R-CHOP in patients with either DLBCL or grade 3b FL (Table 3) [91]. Similarly, a durvalumab (another PD-L1 inhibitor) and R-CHOP combination did not seem to add any benefit (Table 3) [92]. An ongoing phase 3 trial is currently investigating nivolumab combined with DA-EPOCH-R versus DA-EPOCH-R in newly diagnosed patients with PMCBL (NCT04759586).

Relapse/refractory setting

Various therapeutic combinations, including PD-1/PD-L1 ICB, have been investigated in the R/R NHL setting, most of them showing disappointing results. Nivolumab with ipilimumab or lirilumab did not show any improved clinical activity (Table 3) [19,81], and nor did combinations of either nivolumab, pembrolizumab or durvalumab with BTKi (ibrutinib [100,108] or acalabrutinib [96]) or dinaciclib, a cycline kinase inhibitor (CDKi) [107] (Table 3). Similarly, durvalumab in combination with ibrutininb, lenalidomide +/− rituximab or bendamustine +/− rituximab only adds a small benefit [60].

PD-1/PD-L1 ICB combined with CD19 CAR T cells achieved variable results in early-phase trials (Table 3) [58,94,95,109]. Dual-targeting CD19/CD22 CAR T-cell therapy associated with pembrolizumab was evaluated in 19 R/R B-cell NHL patients, with an ORR of 65% with a 55% CMR (Table 3) [97].

Other combined regimens of PD-1/PD-L1 ICB with either other ICBs, targeted agents (tazemetostat [93], venetoclax [104]), novel anti-CD 20 and anti-CD27 antibodies (obinutuzumab [101], varlimumab (NCT03038672)) or CD20/CD3 bispecific antibodies (glofitamab [106], monetuzumab) (NCT02500407)) are under investigation in this setting (Table 3).

In R/R PMBCL patients, the phase 2 CheckMate 436 trial (2019, 2021) evaluated nivolumab with BV (n = 30), reporting an ORR of 73% and a 37% CMR. Responses were durable at extended follow-up with a 2-year PFS of 56% (Table 3) [98,99]. A phase 1 study testing the addition of nivolumab and lenalidomide to immunochemotherapy in PCNSL is now recruiting (NCT04609046). In advanced ENKTL, a combined sintilimab, pegasparaginase, gemcitabine and oxaliplatin regimen has demonstrated an acceptable toxicity profile and promising efficacy in a phase 1 study [119]; a phase 2 trial is currently ongoing (NCT04127227).

With the exceptions of specific NHL subtypes and potentially CAR T-cell association, PD-1/PD-L1 ICBs generally do not add much clinical benefit when combined with other active agents in the R/R NHL setting.

##### Indolent B NHL

Frontline setting

The combination of PD-1/PD-L1 ICB with anti-CD20 monoclonal antibodies +/− bendamustin has been tested on advanced-grade 1-3A FL [46,47]; the addition of chemotherapy resulted in an inacceptable toxicity profile, with fatal adverse events occurring in five patients (pneumonia, sudden death, cardiac arrest (due to severe immune-mediated myocarditis and bronchiolitis obliterans), gastrointestinal tract/biliary adenocarcinoma, progressive multifocal leukoencephalopathy) [83,84] (Table 3).

Relapse/refractory setting

The addition of PD-1/PD-L1 ICB to anti-CD 20 monoclonal antibodies has been extensively tested in the R/R setting of indolent lymphomas, with no or only modest results (Table 3) [85,86,87,101]. Atezolizumab with obinutuzumab and lenalidomide resulted in a 78% ORR and a 72% CR rate [88]. The LYSA group investigated atezolizumab, obinutuzumab and venetoclax in R/R FL and MZL with reported ORRs of 54% and 67%, respectively [105]. Durvalumab or nivolumab in combination with ibrutinib resulted in modest response rates in R/R FL patients in two trials (Table 3) [102,103,108].

## 3. Predictive Biomarkers of Response to ICB in Lymphoma

Predictive biomarkers of response to ICBs have been identified in different solid tumors, but our ability to accurately predict response in lymphoma remains suboptimal. Mechanisms of immune evasion may differ from one lymphoma subtype to another. Several biomarkers have been investigated in lymphoid malignancies, including the PD-L1 H-score, alterations/amplification of the 9p24.1 gene, microsatellite instability (MSI), tumor mutational burden (TMB), the density of intratumoral CD8+ T lymphocyte infiltrates, genetic alterations in MHC classes I and II, and miRNA-21, among others [120].

### 3.1. Tissue Expression and Plasma Levels of PD-Ls

PD-L1 and PD-L2 expression of TME can be influenced by two main signaling pathways. The extrinsic pathway relies on the release of inflammatory signals (i.e., IFN-γ) by TILs after tumor antigen recognition, consequently upregulating the expression of PD-L1/PD-L2 in tumor cells and TME cells [121]. On the other hand, the intrinsic pathway is mainly driven by genomic alteration of the 9p24.1 gene, EBV infection and the activation of the JAK/STAT transcription pathway; this is reviewed below [13,122].

#### 3.1.1. PD-L1 Expression

Different scoring systems are used to determine PD-L1 expression (measured using immunohistochemistry; IHC) in tumors; however, its discriminative value is likely imperfect as patients with PD-L1-negative tumors may achieve durable responses [123]. Additionally, the PD-L1-positive cut-off threshold remains highly variable depending on tumor type, with different scoring systems used to determine PD-L1 expression in tumors [124]. The main scores are the combined positive score (CPS) [125], the tumor proportion score (TPS) [126] and the H-score [120]; the latter is the most-used scoring system for lymphoma [120].

Even though some studies revealed that TME PD-L1 expression correlates with poor prognosis in HL and NHL [127], this does not seem to be true for all lymphoma subtypes (i.e., NK/T lymphoma) [128]. Similarly, PD-L1 expression is not a reliable biomarker of response to PD-1/PD-L1 ICB for all lymphomas. For reasons explained below (refer to section), PD-L1 expression correlates to the response to anti-PD-1 therapy in cHL and some non-GCB DLBCL subtypes, but has no predictive value for other subtypes [128,129]. Similarly, soluble PD-L1 (sPD-L1), corresponding to plasma levels of PD-L1, has been investigated as a potential biomarker, with mixed results [130,131,132].

#### 3.1.2. PD-L2 Expression

The prognostic significance of PD-L2 expression in malignant tumors remains controversial [133,134]. Gu and colleagues demonstrated that PD-L2 expression on DLBCL cells was significantly associated with prolonged OS and PFS. Higher ORRs to R-CHOP/CHOP were also reported in patients with enhanced expression of PD-L2 in malignant and immune cells [135]. Similarly, Tobin et al. demonstrated that PD-L2 is mostly present in the TME of FL and that a low PD-L2 expression was associated with worse outcomes [136].

### 3.2. Gene Alterations to 9p24.1

Copy number alterations (CNAs) to chromosome 9p24.1 (i.e., polysomy, copy gain, amplification, rarely translocation), leading to increased expression of PD-1 ligands in cHL, PMBCL and some extranodal large B-cell NHLs, are an important mechanism of tumor immune evasion. These alterations have been reported in the majority of cHLs (97%) [135], 63–75% of PMBCLs [137,138], over 40% of PTLs and roughly 20% of PCNSLs [139]. Unlike cHL, there is generally a low incidence (10–27%) of structural variations in PD-L1/PD-L2 in DLBCL [140,141]. The H-score could be used as a surrogate marker of the level of 9p24.1 gene alteration and predict the response to ICB [142]. cHL patients presenting 9p24.1 CNAs are more likely to present with advanced-stage disease and worse prognosis, with significantly shorter PFS [61]. On the other hand, they also tend to benefit from therapy with a PD-1 inhibitor [61,135,143]. However, Green et al. (2010) described that Hodgkin Reed–Sternberg cell lines harboring low levels of 9p24.1 CNAs still expressed PD-L1, suggesting that PD-L1 expression could be driven by other mechanisms [135]. Although the vast majority of NHLs have a considerably low sensitivity to PD-1/PD-L1 ICB, particular subtypes, such as PCNSL, PTL and PMBCL harbor 9p24.1 CNAs, conferring them an increased vulnerability to these agents [15,139]. In PMBCL, the magnitude of 9p24.1 CNAs is significantly associated with PD-L1 expression and survival outcome [14].

### 3.3. Epstein–Barr Virus (EBV) and JAK/STAT Signaling Pathway

In cHL and PMBCL cell lines, Green et al. (2010) observed that broader 9p24.1 amplifications also included the *Janus kinase 2* (*JAK2*) locus located upstream from *PD-1 ligand* genes [135]. As a consequence, 9p24.1 CNAs directly enhanced the JAK/STAT intrinsic signaling pathway, promoting *PD-1 ligand* transcription [135]. Additionally, the JAK/STAT signaling pathway can also be activated by various cytokines secreted by cells within the TME (i.e., IFN-γ), leading to PD-L1 upregulation on tumor cells [144].

Even though the underlying mechanisms have not yet been fully elucidated, recent studies indicate that viruses also use the PD-1 signaling pathway to escape immune detection [145]. Thus, EBV-positive lymphomas tend to benefit from ICB therapy [146,147]. The expression of EBV latent membrane protein 1 (LMP1) or latent membrane protein 2a (LMP2a) was shown to be sufficient to activate the signaling cascade of the *JAK/STAT* pathway, leading to PD-L1 overexpression on tumor cells [145]. The expression of EBV has been implicated in approximately 40% of cHLs, 50–70% of post-transplant lymphoproliferative disorders (PTLDs), more than 95% of endemic Burkitt lymphomas (BLs), 20–30% of sporadic BLs, 25–40% of immunodeficiency-associated BLs, EBV-positive DLBCL NOS and most NK/T-cell lymphomas [148,149,150,151]. In a cohort of 1253 patients with DLBCL, PD-L1 protein expression was significantly associated with EBV positivity in the non-GCB subtype and showed a trend toward inferior OS in these patients [152].

### 3.4. Tissue Tumor Mutational Burden (TMB) and Plasma Tumor Mutational Burden (pTMB)

Based on the concept that higher mutation loads could yield to improved T-cell recognition of tumors via increased neoantigen production, tumor mutational burden (TMB), defined as the number of somatic mutations per megabase (mut/Mb) [153], was investigated as a predictive biomarker of response to PD-1/PD-L1 ICB in several solid malignancies [154,155,156]. Surprisingly, TMB has only been reported in a few lymphoma patient series, and its relationship to response to PD-1/PD-L1 ICB is not well established. Using a 406-DNA gene panel, Galanina and colleagues showed a median TMB of 1.7 mut/Mb across all hematologic malignancies [157]. Applying whole-exome sequencing (WES), Wienand et al. described a median TMB of 7.7 mut/Mb in cHL, although TMB may differ per EBV status, with EBV-negative cHL harboring a higher mutational level, almost similar to NSCLC (median 9.8 mut/Mb) and melanoma (13.5 mt/Mb) TMB [158,159,160]. The median TMB of PMBCL has been shown to be within the same range as cHL (7.0 mut/Mb) [161]. Using LymphomaScan, a 405-gene panel, Cho et al. (2021) investigated TMB in different NHLs (B-cell neoplasms n = 243, T- and NK-cell neoplasms n = 53, precursor lymphoid neoplasms n = 4), reporting single-nucleotide variant (SNV) and insertion–deletion mutations (Indels) for each subtype. Overall, they found that B-cell lymphomas had statistically more mutations (24 SNV/Indel) than T- and NK-cell lymphomas (17 SNV/Indel). PMBCL accounted for the highest TMB load (32 SNV/Indel), followed by PCNSL (30 SNV/Indel), DLBCL NOS (23 SNV/Indel), ALK-negative anaplastic large cell lymphoma (ALCL) (23 SNV/Indel), ALK-positive ALCL (14 SNV/Indel), follicular T-cell lymphoma (14 SNV/Indel) and nodal peripheral T-cell lymphoma with TFH phenotype (PTCL TFH) (14.5 SNV/Indel) [162].

Circulating tumor DNA (ctDNA), sometimes referred to as liquid biopsy, reflects tumor DNA spread within the circulation [163]. Plasma TMB (pTMB) has been investigated in solid tumors and DLBCL as a surrogate to quantify overall tumor burden and mutational load and more accurately capture molecular tumor heterogeneity and clonal evolution [164,165]. To our knowledge, only a limited number of clinical studies have been conducted on lymphoma. One study evaluated pTMB in cHL, showing that higher baseline ctDNA and a sharper ctDNA decrease (>40%) significantly correlated with better clinical responses to sintilimab [166]. Another study, reported as an abstract only, compared TMB and pTMB in several NHL subtypes and concluded that there is a higher TMB in DLBCL than in other lymphoma subtypes [167]. Based on these results, it seems that a higher TMB seems to correlate to PD-1 ICB response, as this is the case for other solid cancers [168].

### 3.5. MSI and d-MMR

Microsatellite instability (MSI), caused by deficiency of the DNA mismatch repair (MMR) system, results in a higher mutational load and tumor antigen expression, leading to increased antitumor T-cell activation. MSI and MMR status are well-recognized prognostic factors in several solid tumors. The National Cancer Institute recommends a panel of five microsatellite markers for MSI tumor detection (two mononucleotide repeats: BAT-25, BAT-26; three dinucleotide repeats: D2S123, D5S346, D17S250). MSI-high tumors are defined as having instability in two or more of these markers [169]. Like those harboring a high TMB, patients with MSI-high tumors achieve durable responses to ICB [170]. Pembrolizumab and dostarlimab, both anti-PD-1 antibodies, are now approved by the FDA for the treatment of unselected advanced-stage tumors with MSI/dMMR [171]. Even though MSI-associated hypermutation represents a potential biomarker for the efficacy of PD-1 blockade, it is likely infrequent in lymphoma; reported frequencies are low, occurring in only 0.46% of cHLs, 3.2% of DLBCLs and 8% of PMBCLs [160,161,172].

### 3.6. MHC Expression

Antigen presentation by major histocompatibility complex (MHC) molecules is an essential step in T-cell recognition and tumor cell eradication, as T-cell activation does not occur at the tumor site but in the lymph nodes. The MHC I complex, including the beta-2-microglobulin (B2M) subunit, presents tumor-generated peptides at the cell surface, which can be recognized by cytotoxic CD8+ T cells. On the other hand, MHC II complexes present tumor antigens to CD4+ T cells [173]. Acquired mutations in the antigen processing and presentation molecules are a potential mechanism of tumor escape [174]. B2M mutations resulting in decreased MHC I expression are a well-described acquired resistance mechanism to PD-1/PD-L1 ICB in melanoma [175]. By contrast, in a study on cHL, Roemer et al. (2016) discovered that over 75% of patients had decreased or absent expression of B2M/MHCI and MHCII on tumor cells [176]. Nonetheless, patients with cHL are known to have a high response rate to PD-1 ICB. The postulated alternate mechanism triggering a response to PD-1/PD-L1 ICB could be MHC class II expression. Indeed, melanoma patients with low or absent B2M/MHCI expression but increased MHCII expression demonstrated higher responses to ICB [177]. This suggests that MHCII expression could be a potential biomarker for PD-1/PD-L1 ICB response and requires further evaluation.

### 3.7. Intratumoral CD8+ T Lymphocyte Infiltrate Density

Several studies have demonstrated that response to PD-1/PD-L1 ICB seems tightly related to immune cell infiltration. High levels of cytotoxic CD8+ T lymphocytes at the TME have been independently linked to improved outcomes in several lymphomas (i.e., DLBCL [178], PTL [179], FL [180], MZL [181] and HL [182,183]). The association between tumor CD8+ T lymphocyte infiltration and response to PD-1/PD-L1 ICB has been confirmed for solid tumors through a large metanalysis conducted by Li et al. (2021) [184]. Such data have not yet been reported for lymphoma.

### 3.8. MicroRNAs

MicroRNAs (miRNAs) are small sequences of noncoding RNAs acting as gene expression regulators. miRNAs are involved in various physiologic and pathologic processes, including immune responses [185]. Numerous miRNAs are recognized as prognostic biomarkers and are investigated as potential therapeutic targets in solid and hematologic tumors [186]. Among all miRNAs, microRNA-21 (miR-21) is one of the most frequently overexpressed miRNAs in solid tumors and B-NHL [187]. High plasma miR-21 levels have been associated with poor prognosis in several B-NHL subtypes (e.g., primary gastrointestinal DLBCL [188], DLBCL [189], Burkitt lymphoma [190], PCNSL [191]). miR-21 depletion may enhance antitumor activity through the polarization of macrophages into an M1-like phenotype; on the other hand, miR-21 also upregulates PD-L1 expression [192]. Taking advantage of these properties, Xi et al. recently showed in a preclinical model a synergetic effect of miR-21-depleting therapy and PD-1 ICB [192].

### 3.9. Gut Microbiome

Recent evidence shows that the biodiversity of the gut microbiome could influence the antitumor activity of PD-1/PD-L1 ICB; this was notably investigated in melanoma [193,194]. Namely, Liu et al. (2021) demonstrated that a favorable gut microbiome characterized by its diversity and the presence of specific bacteria species could influence the innate and adaptive immune system by increasing antigen presentation and augmenting T-cell response. On the other hand, antibiotic use may disrupt the gut microbiome and impair cytotoxic T-cell responses against tumor cells [195]. Hwang et al. studied this association in a small retrospective cohort of 62 cHL patients treated with ICB and observed that prior and/or current antibiotherapy was linked to inferior outcomes [196]. Recently, Casadei et al. (2021) prospectively collected feces (at baseline, before each treatment, at response assessment and for grade >2 adverse events) from cHL (n = 12) and PMBCL (n = 5) patients undergoing PD-1PD-L1 ICB. They reported that the results of the first six patients (all cHL) showed clear differences in their microbiomes, with a depletion of health-promoting microbial components compared with healthy controls [197].

## 4. Conclusions

PD-1/PD-L1 ICB is now a well-recognized therapy against specific lymphoproliferative disorders such as cHL and PMBCL. Besides those specific indications, PD-1/PD-L1 ICB shows generally disappointing results across all NHLs. Some patients with specific NHL subtypes, (e.g., PMBCL, PTL, PCNSL) or particular molecular findings (e.g., high TMB), may derive prolonged clinical benefit from these agents. To date, the identification of discriminative biomarkers of response to PD-1/PD-L1 ICB has been very challenging. In lymphoma, PD-L1 expression is driven by genetic alterations of the 9p24.1 locus, activation of the JAK/STAT signaling pathway and EBV infection. However, its expression is highly variable among lymphoma subtypes and does not always correlate with clinical responses to PD-1/PD-L1 ICB. Additionally, the value of a PD-L1 H-score cut-off is currently not fully understood. Other biomarkers, such as high TMB and high MSI, have not been clearly established for lymphoma, so further research in this area is urgently needed.

## Figures and Tables

**Table 1 biomedicines-11-01720-t001:** Currently approved PD-1 antibodies in lymphoma.

Target	Name	Pharmaceutical Company	Isotype	Approval Status	Year Indication
PD-1	Nivolumab (Opdivo^®^)	Bristol-Myers Squibb	IgG4 S228P	FDA/EMA [19,20]	2018, cHL
	Pembrolizumab (Keytruda^®^)	Merck	IgG4 S228P	FDA/EMA [21,22,23]	2018, cHL, PMBCL
	Tislelizumab (BGB-A317)	BeiGene	IgG4mut, FcyR null	China NMPA [24]	2019, cHL
	Camrelizumab (AiRuiKa™)	Hengrui	IgG4 S228P	China NMPA [25]	2019, cHL
	Sintilimab (Tyvyt^®^)	Innovent Biologics, Eli Lilly	IgG4 κ	China NMPA [26]	2018, cHL
	Zimberelimab (AB122)	Gloria Biosciences, Arcus Biosciences, Taiho Pharmaceutical Co	IgG4	China NMPA [27]	2021, cHL
	Penpulimab (AK105)	Akeso Biopharma	IgG1, FcyR null	China NMPA [28]	2021, cHL

Abbreviations: cHL, classical Hodgkin lymphoma; FDA, Food and Drug Administration; EMA, European Medicines Agency; NMPA, National Medical Products Administration; PMBCL, primary mediastinal B cell lymphoma.

**Table 2 biomedicines-11-01720-t002:** PD-1/PDL-1 ICB monotherapy prospective studies by lymphoma subtype.

Lymphoma Subtype	Studies	N	Study Population	Therapy	ORR (%)	CR (%)	Median PFS (Months)	Median OS (Months)	Median Follow-Up (Months)	Specificities
Classical Hodgkin lymphoma	Ansell 2015, [11,29], Phase 1	23	R/R (prior ASCT or allo-SCT)	Nivolumab 3 mg/kg q2wks *	87	17	NR † (86% at 6 months)	NR †	21.5	
	Maruyama 2017, 2020, [30,31], Phase 2	17	R/R (prior ASCT and BV)	Nivolumab 3 mg/kg q2wks *	88	31	11.7 (60% at 6 months)	NR (80% at 3 years)	38.8	
	KEYNOTE-013, Armand 2016, 2020 et al., [21,32], Phase 2B	31	R/R after BV failure	Pembrolizumab 10 mg/kg q2wks *	58	19	11.4 (30% at 2 years)	NR (81% at 3 years)	52.8	
	CheckMate 205, Younes 2016, Armand 2018, Ansell 2021, [10,20,36,37], Cohorts A, B, C, Phase 2	243	R/R after: A. ASCT (n = 63) B. ASCT + BV (n = 80) C. BV + ASCT ± BV (n = 100)	Nivolumab 3 mg/kg q2wks *	71	21	15 (18% at 5 years)	NR (71% at 5 years)	58	
	KEYNOTE-087, Ansell 2017, Chen 2019, Chen 2021, [12,22,38], Phase 2	210	R/R after: A. ASCT + BV (n = 69) B. Salvage chemo + BV (n = 89) (ineligibility for SCT owing to chemorefractory disease) C. ASCT (n = 60)	Pembrolizumab 200 mg q3wks *	71	28	13.7 (14% at 5 years)	NR (71% at 5 years)	62.9	
	KEYNOTE-204, Kuruvilla 2021, [35], Phase 3	304	R/R (ineligible or relapsed after ASCT)	Pembrolizumab 200 mg q3wks versus BV 1.8 mg/kg q3wks *	Pembrolizumab: 66 BV: 54	Pembrolizumab: 25 BV: 24	13.2 (for pembrolizumab) 8.3 (for BV)	NA	25.7	
	JAVELIN Hodgkin trial, Herrera 2021, [39], Phase 1B	31	R/R (ineligible or relapsed after ASCT)	Avelumab with four dose levels and two dosing schedules (q2wks or q3wks)	42	19	5.7 (18% at 1 year)	NA	NA	Dose levels/schedules: 70, 350 and 500 mg q2wks; 500 mg q3wks; 10 mg/kg q2wks
	Song 2019, Song 2022, [24,40], Phase 2	70	R/R (ineligible or relapsed after ASCT)	Tislelizumab 200 mg q3wks *	87	67	31.5 (41% at 3 years)	NR (85% at 3 years)	33.8	
	Song 2019, Wu 2021, [25,41], Phase 2	75	R/R (ineligible or relapsed after ASCT)	Camrelizumab 200 mg q2wks *	76	28	22.5 (67% at 1 year)	NR (83% at 3 years)	36.2	
	Nie 2019, Liu 2021, [42,43], Cohort 1, Phase 2	19	R/R after more than 2 therapy lines, anti-PD-1 naïve	Camrelizumab 200 mg q3wks monotherapy *	90	32	15.5 (42% at 2 years)	NR (63% at 2 years)	34.5	
	ORIENT-1, Shi 2019, Su 2020, [26,44], Phase 2	96	R/R after more than 2 therapy lines (including ASCT)	Sintilimab 200 mg q2wks *	80	29	18.6 (78% at 6 months)	NR (96% at 2 years)	26.7	
	Armand 2019, [45], Phase 2	30	R/R after ASCT	Pembrolizumab 200 mg q3wks × 8 cycles as maintenance after ASCT	NA	NA	NR (82% at 18 months)	NR (100% at 18 months)	NA	
	Song 2022, [28], Phase 1/2	85	R/R (including ASCT)	Penpulimab 200 mg q2wks * (maximum of 24 months)	89	47	NR (72% at 1 year)	NR (100% at 18 months)	15.8	
	Lin 2022, [27], Phase 2	85	R/R (including ASCT)	Zimberelimab (GLS-010) q2wks * (maximum of 24 months)	91	33	NR (78% at 1 year)	NR (99% at 1 year)	15.8	
Follicular lymphoma	Barraclough 2019, Hawkes 2021, [46,47], Phase 2	39	Newly diagnosed, stage III-IV, grade 1-3a FL	Nivolumab 240 mg 2-weekly × 4 cycles If CR: N 240 mg monotherapy × 4 cycles, maintenance N 480 mg 4-weekly × 12 cycles If < CR: N 240 mg + rituximab 375mg/m^2^ 2-weekly × 4 cycles, maintenance N+R (N 480 mg 4 weekly ×12 cycles; R 12 weekly × 8 cycles).	92	54	NR (72% at 1 year)	NR (96% at 1 year)	17.5	
	CheckMate 140, Armand 2021, [48], Phase 2 R/R	92	R/R (after failure of at least 2 prior lines of therapy)	Nivolumab 3 mg/kg q2wks *	4	1	2.2	NA	NA	12-months minimal follow-up
Diffuse large B-cell lymphoma	Ansell 2019, [49], Phase 2	121	R/R (ineligible or relapsed after ASCT)	Nivolumab 3 mg/kg q2wks *	10 (all groups) ‡	3 (all groups) §	ASCT failed (n = 87): 1.9 ASCT ineligible (n = 34): 1.4	ASCT failed (n = 87): 12.2 ASCT ineligible (n = 34): 5.8	ASCT failed: 9 ASCT ineligible: 6	
Primary mediastinal B-cell lymphoma	KEYNOTE-013, Zinzani 2017, Armand 2019, [14,50] Phase 1B	21	R/R PMBCL	Pembrolizumab 10 mg/kg (n = 11) 200 mg (n = 10) q3wks *	48	33	10.4	31.4	29.1	
	KEYNOTE-170, Armand 2019, [14] Phase 2	53	R/R PMBCL	Pembrolizumab 200 mg q3wks *	45	13	5.5	NR	12.5	
Extranodal natural killer/T-cell lymphomas	Kim 2020, [51], Phase 2	21	R/R ENKTL	Avelumab 10 mg/kg q2wks *	38	24	2.7	NR	15.7	Response significantly associated with the expression of PD-L1 by tumor tissue
	ORIENT-4, Tao 2021, [52], Phase 2	28	R/R ENKTL	Sintilimab 200 mg q3wks *	75	21	NA	NR (79% at 2 years)	30.4	
Hematologic malignancies	Lesokhin 2016, [53], Phase 1	81	R/R B-cell lymphoma, TCL, MM (inclusive after ASCT)	Nivolumab 1 or 3 mg/kg q2wks *	FL (n = 10): 40 DLBCL (n = 11): 36 TCL (n = 23): 17 MM (n = 27): 4	FL (n = 10): 10 DLBCL (n = 11): 18 TCL (n = 23): 0 MM (n = 27): 4	FL (n = 10): NR DLBCL (n = 11): 7 TCL (n = 23): 10 MM (n = 27): 10	NA	16.7	
	Frigault 2020, [54], Phase 2	29	R/R DLBCL + PMBCL after ASCT as maintenance	Pembrolizumab 200 mg q3wks × 8 cycles	N/A	59% at 18 months	NR (59% at 18 months)	NR (93% at 18 months)	NA	
	Davids 2020, [55], Phase 1	28	Relapsed hematologic malignancies after allo-SCT	Nivolumab 0.5–3 mg/kg q2wks *	29	4	3.7	21.4	11	
	Ding 2017, [56], Phase 2	25	Relapsed or progressive CLL (n = 16) + CLL with RT (n = 9)	Pembrolizumab 200 mg q3wks *	CLL: 0 RT: 44	CLL: 0 RT: 11	CLL: 2.4 RT: 5.4	CLL: 4.3 RT: NR	11	
	Khodadoust 2020, [57], Phase 2	24	R/R MF (n = 9) and SS (n = 15)	Pembrolizumab 2 mg/kg q3 wks #	MF: 56 SS: 27	MF: 0 SS: 7	MF: 12 SS: 12	MF: NR SS: NR	NA	
	Chong 2018, [58], Phase 1/2	12	R/R B-cell NHL after CAR T-cell therapy	Pembrolizumab 200 mg q3wks *	27	9	NA	NA	NA	
	iMATRIX, Geoerger 2020, [59], Phase 1/2	90 (n = 18 between 18–29 years)	R/R solid tumors, HL (n = 9), NHL (n = 3)	Atezolizumab 15 mg/kg (<18 years) or 1200 mg (18–29 years) q3wks *	4.6	NA	1.3	7.4	6.8	Patients < 30 years
	FUSION NHL 001 trial, Casulo 2023, [60], Arm D, Phase 1	27	R/R CLL/SLL (n = 2), FL (n = 5), DLBCL (n = 10), MCL (n = 5), HL (n = 5)	Durvalumab 1500 mg q4wks	NR	NR	CLL/SLL: 2.8 FL: 1.7 DLBCL: 1.2 MCL: 2.3 HL: 2.7	CLL/SLL: NR FL: 2.9 DLBCL: 1.6 MCL: 13.6 HL: 23.8	23.3	

* Until progression or unacceptable toxicity; † PFS and OS are data of the first article with the 10-month follow-up. ‡ ORR within groups: ASCT failed (n = 87): 9; ASCT ineligible (n = 34): 1; § CR within groups: ASCT failed (n = 87): 3; ASCT ineligible (n = 34): 0; # until progression or unacceptable toxicity or investigator choice. Abbreviations: allo-SCT, allogeneic stem cell transplantation; ASCT, autologous stem cell transplantation; BV, brentuximab-vedotin; cHL, Hodgkin lymphoma; CLL, chronic lymphocytic leukemia; CMR, complete metabolic response; CR; complete response; DLBCL, diffuse large B-cell lymphoma; ENKTL, extranodal natural killer/T-cell lymphomas; FL, follicular lymphoma; MF, mycosis fungoid; MM, multiple myeloma; N, nivolumab; NA, no data available; NHL, non-Hodgkin lymphoma; NR, not reached; PCNSL, primary central nervous system lymphoma; PMBCL, primary mediastinal B-cell lymphoma; q2wks, every two weeks; q3wks, every three weeks; RT, Richter transformation; SS, Sézary syndrome; TCL, T-cell lymphoma.

**Table 3 biomedicines-11-01720-t003:** PD-1/PDL-1 ICB combination prospective studies by lymphoma subtype.

Lymphoma Subtype	Studies	N	Study Population	Therapy	ORR (%)	CR (%)	Median PFS (Months)	Median OS (Months)	Median Follow-up (Months)	Specificities
Classical Hodgkin lymphoma	CheckMate 205, Ramchandren 2019, [61] Cohort D, Phase 2	51	Untreated advanced-stage	Nivolumab 240 mg q2wks × 4 doses followed by nivolumab-AVD × 6 cycles	84	67	NR (92% at 9 months)	NR (98% at 9 months)	11.1	
	NIVAHL Trial, Bröckelmann 2022, [62], Phase 2	109	Untreated early-stage and unfavorable	Group 1: nivolumab 240 mg q2wks + AVD × 4 cycles Group 2: nivolumab × 4 cycles monotherapy, nivolumab-AVD × 2 cycles, AVD only × 2 cycles, followed by 30 Gy involved-site radiotherapy	96	87	NA 1: 100% at 1 year 2: 98% at 1 year	NA 1: 100% at 1 year 2: 100% at 1 year	13	
	Cheson 2020, [67] Phase 2	46	Untreated, >60 years old or younger and ineligible for chemotherapy	Nivolumab 3 mg/kg + BV 1.8 mg/kg q3wks × 8 cycles	61	48	18.3	NR	21.2	
	Allen 2021, [63], Phase 2	30	Untreated early unfavorable and advanced-stage	Pembrolizumab 200 mg q3wks for 3 cycles, AVD × 4–6 cycles	100	NA	NR	NR	22.5	
	Nie 2019, Liu 2021 et al., [42,43], Cohort 1 combination, Phase 2	67	R/R after more than 2 therapy lines	Cohort 1 combination; anti-PD-1 naïve: (n = 42): decitabine (10 mg/d, days 1 to 5) plus camrelizumab 200mg q3wks * Cohort 2 (n = 25); anti-PD-1 resistant: decitabine plus camrelizumab *	1: 95 2: 52	1: 71 2: 28	1: 89% at 1 year 2: 59% at 1 year	1: 63% at 2 years2: NA	34.5	
	Herrera 2019, Mei 2022, [68,69], Phase 2	43	R/R first salvage therapy and bridge to ASCT	Nivolumab 3 mg/kg q2wks × 6 cycles +/− ICE. PET-CT after C3 and C6. After C6, pts in CR: ASCT, not in CR: N-ICE for 2 cycles	93	91	NA (72% at 2 years)	NA (95% at 2 years)	NA	Among 9 patients who received N-ICE: ORR 100%, CR 89%
	Bryan 2021, [70], Phase 2	42	R/R prior to ASCT	Pembrolizumab 200mg q2wks + ICE × 2 cycles, stem cell mobilization/collection, pembrolizumab 200 mg × 1 cycle	97	NA	26.9 (88% at 2 years)	NR (95% at 2 years)	27	
	Diefenbach 2020, [71], Phase 1/2	64	R/R	BV 1.8 mg/kg + ipilimumab 3 mg/kg or nivolumab 3 mg/kg or nivolumab 3 mg/kg and ipilimumab 1 mg/kg *	BV + Ipi76 BV + Nivo89 BV + Ipi/Nivo82	BV + Ipi57BV + Nivo61BV + Ipi/Nivo73	BV + Ipi14.4 (61% at 1 year)BV + NivoNR (70% at 1 year)BV + Ipi/NivoNR (80% at 1 year)	NR	BV + Ipi: 31.2 BV + Nivo: 28.8 BV + Ipi/Nivo: 20.4	
	Sermer 2020, Sermer 2021, [72,73], Phase 2	22	R/R (heavily pretreated, previous ICB therapy accepted)	Entinostat 5–7 mg orally q1wks + pembrolizumab 200mg q3wks	86	45	NA (72% at 1 year)	NA	8.4	
	Lepik 2020, [74], Phase 2	30	R/R after nivolumab monotherapy	Nivolumab 3 mg/kg on D1, 14 + bendamustine (90 mg/m^2^) on D1, 2 of a 28-day cycle for up to 3 cycles	87	57	10.2 (23% at 2 years)	NA (97% at 2 years)	25	
	Herrera 2018, Advani 2021, [75,76] Phase 1/2	62	R/R in initial salvage therapy before ASCT	BV + nivolumab 3 mg/kg q3wks × 4 cycles	82	61	NR (77% at 3 years)	NR (93% at 3 years)	34.3	
	Moskowitz 2021, [77], Phase 2	38	R/R after first-line therapy, prior to ASCT	Pembrolizumab 200 mg + GVD q3wks × 2–4 cycles	100	95	NA	NA	13.5	
	Ansell 2019, Bartlett 2020, [78,79], Phase 1B	24	R/R, CD30-positive, 3–7 prior lines of therapy including BV	Pembrolizumab 200 mg q3wks + AFM13 dose escalation schedules	83	37	NA (77% at 6 months)	NA	NA	
	Dave 2022, [80], Phase 1	10	R/R inclusive after ASCT, allo-SCT, BV, prior ICB) (n = 8 active disease, n = 2 adjuvant after ASCT)	TAA-Ts + nivolumab in 6 patients *		1 CR 7 SD at 3 months			41 (adjuvant arm) and 12.6 (active disease arm)	Nivolumab priming impacted TAA-T recognition and persistence.
	CheckMate 039, Ansell 2016, Armand 2021, [19,81], Phase 1B	52	R/R after ≥2 prior lines of therapy, independent of ASCT	Nivolumab 3 mg/kg + ipilimumab 1 mg/kg q3wks or nivolumab 3 mg/kg + lirilumab 3 mg/kg q4wks *	Nivo/Ipi: (n = 31) 74 Nivo/Liri (n = 21): 76	Nivo/Ipi: (n = 31) 23 Nivo/Liri: (n = 21):24	Nivo/Ipi: NR Nivo/Liri: NR	NA	Nivo/Ipi: 18 Nivo/Liri: 11	
	Timmerman 2022, [82], Phase I/II (cohort 2)	29	R/R after anti-PD-1 therapy	Favezelimab 800mg q3wks + pembrolizumab 20mmg q3wks for up to 35 cycles	31	7	9 (39% at 1 year)	26 (91% at 1 year)	N/A	
Follicular lymphoma	Younes 2017, Younes 2022, [83,84] Phase 1/2	40	FL grade 1, 2 or 3a disease requiring therapy	Obinutuzumab 1000 mg on days 1, 8 and 15 of cycle 1 and day 1 of cycles 2–6, bendamustine 90 mg/m^2^ on days 1 and 2 of cycles 1–6 and atezolizumab 840 mg on days 1 and 15 of cycles 2–6 (28-day cycles). Maintenance in pts with CR or PR consisted of obinu 1000 mg on day 1 of every other month and atezo 840 mg on days 1 and 2 of each month *	NA	75%	NA (81% at 3 years)	NA (89% at 3 years)	40.4	Grade 5 (fatal) adverse events reported in five patients
	Westin 2014, [85], Phase 2	30	Relapsed FL rituximab sensible	Rituximab 375 mg/m^2^ weekly for 1 cycle + pidilizumab 3 mg/kg q4wks for 12 doses	66	52	18.8	NA	15.4	
	Nastoupil 2017, Nastoupil 2022, [86,87], Phase 2	30	Relapsed FL rituximab sensible	Pembrolizumab 200 mg q3wks for up to 16 cycles + rituximab 375 mg/m^2^ weekly for 1 cycle	67	50	12.6	NR (97% at 3 years)	35	
	Morschhauser 2021, [88], Phase 1B/2	32	R/R FL (grade 1–3a)	Obinutuzumab 1000 mg + atezolizumab 840 mg + lenalidomide 15mg (in the expansion phase) or 20 mg × 6 cycles, if CR/PR/SD maintenance*	78 (at the end of the induction)	72 (at the end of the induction)	NA (68% at 3 years)	NA (90% at 3 years)	30 (lenalidomide 15 mg) 14.2 (lenalidomide 20 mg)	
Diffuse large B-cell lymphoma	Younes 2018, Younes 2019, [89,90], Phase 1/2	40	Untreated advanced DLBCL	Atezolizumab 1200mg q3wks + R-CHOP × 8 cycles	NA	78	NA (75% at 2 years)	NA (86% at 2 years)	21.3	
	Smith 2020, [91], Phase 1	30	Untreated DLBCL or grade 3b FL	Pembrolizumab 200 mg + R-CHOP q3wks × 6 cycles	90	77	NA (83% at 2 years)	NA (84% at 2 years)	25.5	
	Nowakowski 2022, [92], Phase 2	37	High-risk DLBCL (IPI ≥ 3/NCCN-IPI ≥ 4)	Durvalumab 1125mg q3wks+ R-CHOP × 6–8 cycles, then durvalumab consolidation *	97	68	NA (68% at 1 year)	NA	NA	
	Palomba 2022, [93], Phase 1B	43	R/R DLBCL	Atezolizumab 1200 mg + tazemetostat 800 mg orally twice daily q3wks	16	7	2	13	23.7	
	ZUMA 6, Jacobson 2020, [94], Phase 1/2	28	R/R DLBCL	Atezolizumab 1200mg + KTE-C19 (axi-cel)	75	46	NR	NR	10.2	
	PORTIA trial, Jäger 2021, [95], Phase 1B	12	R/R DLBCL	Pembrolizumab q3wks for up to 6 doses either days +15, +8 or –1 of tisagenlecleucel	Days + 15: 50 Days + 825 Days-1 25	Days + 15: 0 Days + 8 25 Days-1 25	NA	NA	4	
	Witzig 2019, [96], Phase 1/2	61	R/R DLBCL	Pembrolizumab 200 mg q3wks + acalabrutinib 100 mg BID *	26	7	1.9	NA	NA	
	Alexander trial, Osborne 2020, [97], Phase 1	29	R/R DLBCL	Pembrolizumab 200 mg q3wks + AUTO3 (bispecific CAR T targeting CD19/22)	69	52	NA	NA	NA	
Primary mediastinal B-cell lymphoma	CheckMate 436, Zinzani 2019, Zinzani 2021, [98,99], Phase 2	30	R/R PMBCL	Nivolumab 240 mg and BV 1.8 mg/kg q3wks *	73	37	26 (56% at 1 and 2 years)	NR (76% at 2 years)	11.1	
Multiple hematologic malignancies	CheckMate 039, Ansell 2016, Armand 2021, [19,81], Phase 1B	78	R/R hematologic malignancies (≥2 prior lines of therapy) independent of ASCT	Nivolumab 3 mg/kg + ipilimumab 1 mg/kg q3wks or nivolumab 3 mg/kg + lirilumab 3 mg/kg q4wks *	N/I: B-NHL (n = 16) 19 T-NHL (n = 11): 9 N/L B-NHL (n = 32): 13 T-NHL (n = 11): 22	N/I: B-NHL(n = 16)6 T-NHL (n = 11): 0 N/L B-NHL (n = 32):3 T-NHL (n = 11): 0	N/I: B-NHL(n = 16) 1 T-NHL (n = 11): 2 N/L B-NHL (n = 32): 1 T-NHL (n = 11): 6	NA	N/I: 18 N/L: 11	
	Younes 2019, [100], Phase 1/2A	141	R/R CLL, SLL, FL, DLBCL	Ibrutinib + nivolumab 3 mg/kg q2wks *	CLL/SLL (n = 36):61, FL (n = 40):33, DLBCL (n = 45):36RT (n = 20):65	CLL/SLL (n = 36):0 FL (n = 40):10DLBCL (n = 45):16 RT (n = 20): 10	CLL/SLL (n = 36): NR FL (n = 40):9.1 DLBCL (n = 45):2.6RT (n = 20): 5.0	CLL/SLL (n = 36): NR FL (n = 40):NR DLBCL (n = 45):13.5RT (n = 20): 10.3	19.7	
	Palomba 2022, [101], Phase 1B	49	R/R FL + R/R DLBCL	Atezolizumab 1200 mg + obinutuzumab 1000 mg	FL (n = 26): 54 DLBCL (n = 23): 17	FL (n = 26): 23 DLBCL (n = 23): 4	FL (n = 26): 9 DLBCL (n = 23): 3	FL (n = 26): NA DLBCL (n = 23): 9	NA	
	Jain 2016, Jain 2018, [102,103], Cohort 1, Phase 2	28	R/R FL + RT	Nivolumab 3 mg/kg q2wks × 24 cycles + ibrutinib 420 mg *	FL (n = 5): 60 RT (n = 23): 43	FL (n = 5): 0 RT (n = 23): 35	FL (n = 5): NR RT (n = 23): NA	FL (n = 5): NR RT (n = 23): 13.8	NA	
	LYSA trial, Herbaux 2020, Herbaux 2021, [104,105], Phase 2	136	R/R DLBCL (cohort 1) and R/R iNHL (FL + MZL) (cohort 2)	Obinutuzumab 1 g × 8 cycles + atezolizumab 1.2 g q3wks × 24 cycles + venetoclax 800 mg/day (on D8) × 24 cycles	DLBCL (n = 58): 24 FL (n = 58): 54 MZL (n = 20): 67	DLBCL (n = 58): 18 FL (n = 58): 30 MZL (n = 20): 17	NA	NA	DLBCL: 9FL: 14.5 MZL: 11.9	
	Hutchings 2019, [106], Phase 1B	36	R/R B- NHL (DLBCL, transformed FL, MCL, PMBCL, LPL, iNHL)	Atezolizumab 1200 mg + CD-20-TCB antibody (RG6026) q3wks	36	17	NA	NA	NA	Ongoing trial
	KEYNOTE-155, Gregory 2022, [107], Phase 1B	72	R/R hematologic malignancies	Pembrolizumab 200 q3wks + dinaciclib (dose escalation) *	CLL (n = 17): 29 DLBCL (n = 38): 21 MM (n = 17): 0	CLL (n = 17): 0 DLBCL (n = 38): 11MM (n = 17): 0	CLL (n = 17): 5.2 DLBCL (n = 38): 2.1 MM (n = 17): 1.6	CLL (n = 17): 21.7 DLBCL (n = 38): 7.9 MM (n = 17): 10.5	NA	Dinaciclib dose levels (7 mg/m^2^, 10 mg/m^2^, 14mg/m^2^)
	Herrera 2020, [108], Phase 1/2	61	R/R FL + R/R DLBCL	Durvalumab 10 mg/kg q2wks + ibrutinib 560 mg once daily (dosing according to phase 1B) *	FL (n = 29): 26 Non-GCB DLBCL (n = 16): 38 GCB-DLBCL (n = 16): 13	FL (n = 29): 4 Non-GCB DLBCL (n = 16): 31 GCB-DLBCL (n = 16): 6	FL (n = 29): 10.2 Non-GCB DLBCL (n = 16): 4.1 GCB-DLBCL (n = 16): 2.9	FL (n = 29): NR Non-GCB DLBCL (n = 16): 7.3 GCB-DLBCL (n = 16): 5.5	FL: 17 DLBCL: 17.5	
	Hirayama 2020, [109], Phase 1	13	R/R B-cell NHL	Durvalumab dose escalation up to 10 doses + JCAR014 (CD19-specific 4-1BB-costimulated CAR T cells)	50	42	NA	NA	NA	Durvalumab dose escalation ongoing
	Panayiotidis 2022, [110], Phase 2	55	R/R MCL, WM, MZL	Atezolizumab + obinutuzumab (MCL + WM) or rituximab (MZL)	MCL (n = 30): 17 WM (n = 4): 0 MZL (n = 21): 43	NA	NA	NA	NA	
	Sang 2022, [111], Phase 2	12	R/R CD30+ lymphoma (9 cHL, 1 angioimmunoblastic T-cell lymphoma (AITL), 2 gray zone lymphoma)	Cohort 1: 10^6^/kg of CD30 CAR Ts Cohort 2: 10^7^/kg of CD30 CAR Ts Cohort 3: 10^7^/kg of CD30 CAR Ts + anti-PD-1 antibody q3wks starting 14 days after CAR T-cell infusion †	92 (Cohorts 1 and 2: 86%; Cohort 3: 100%)	70 (Cohorts 1 and 2: 27%; Cohort 3: 80%)	45	70	21.5	Anti-PD-1 treatment not mentioned
	FUSION NHL 001 trial, Casulo 2023, [60], Arm A, B, C, Phase 1/2	34	Arm A: B-cell NHL (n = 14) Arm B: B-cell NHL/CLL (n = 7) Arm C: B-cell NHL/CLL (n = 13)	Durvalumab 1500mg q4wks + Arm A: Len 20mg, Len 20 or 10 mg +/− R Arm B: Ibr 420 (CLL/SLL) or 560mg (MCL) Arm C: Ben 70mg + R (FL, DLBCL, CLL/SLL)	Arm A: 66.7, 66.7, 80 Arm B: 100, 70 Arm C: 88.9, 30, 50	Arm A: 33.3, 33.3, 20 Arm B: 0, 30 Arm C: 44.4, 10, 25	Arm A: 8.4, NA, NA Arm B: CLL/SLL: NA MCL: NA Arm C: FL: 14.7 DLBCL 2.1 CLL/SLL: NA	Arm A: NA, NA, NA Arm B: CLL/SLL: NA MCL: NA Arm C:FL: NA DLBCL 5.1 CLL/SLL: NA	Arm A: 23Arm B: 23.3 Arm C: 14.8	Arm A: no dose-confirmation cohort

* Until progression or unacceptable toxicity. † Sequential HSCT was performed in part on patients; second infusion of CD30 CAR Ts allowed if PD after first treatment or after ASCT. Abbreviations: ASCT, autologous stem cell transplantation; doxorubicin, vinblastine, dacarbazine; Ben, bendamustine; CHOP, cyclophosphamide, hydroxyadriamycine, vincristine, prednisone; CR, complete remission; DLBCL, diffuse large B-cell lymphoma; FL, follicular lymphoma; GVD, gemcitabine, vinorelbine, liposomal doxorubicin; HL, Hodgkin lymphoma; HSCT, hematopoietic stem cell transplantation; Ibr, ibrutinib; ICE, ifosfamide, carboplatine, etoposide; iNHL, indolent non-Hodgkin lymphoma; I, ipilimumab; Len, lenalidomide; L, lirilumab; MCL, mantle cell lymphoma; MZL, marginal zone lymphoma; N, nivolumab; NA, no data available; NR, not reached; PD, progressive disease; PMBCL, primary mediastinal B-cell lymphoma; PR, partial response; q2wks, every two weeks; q3wks, every three weeks; R, rituximab; TAA-Ts, tumor-associated antigen-specific T cells; TCB, T-cell-engaging bispecific; WM, Waldenström’s macroglobulinemia.

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
