# Peer review of "Efficacy of Immune Checkpoint Blockade and Biomarkers of Response in Lymphoma: A Narrative Review"

_biomedicines, 2023, doi:10.3390/biomedicines11061720_

Round 1

Reviewer 1 Report

This review covers Immune checkpoint blockade and biomarkers of response in lymphoma. This is an excellent review. In fact, it is as good as any review that I have examined on any subject.

The inclusion of abbreviations for each section is of great help.

It is clear that the authors have a significant knowledge in this field, and have used this knowledge to compose an excellent reference that should be used for any publication on this topic. 

Figures 1 and 2 are well represented and explained. I have no problems with them

The 209 references are up-to-date and very well represent this field, and provide a wealth of information on this topic. I anticipate that this review will be used as a reference in any future publication on this topic.

 I highly recommend acceptance of this review, and look forward to seeing it as a publication.

This is one of the rare occasions when I see a manuscript so complete and well represented.

Author Response

REVIEWER 1:

This review covers Immune checkpoint blockade and biomarkers of response in lymphoma. This is an excellent review. In fact, it is as good as any review that I have examined on any subject.

The inclusion of abbreviations for each section is of great help.

It is clear that the authors have a significant knowledge in this field, and have used this knowledge to compose an excellent reference that should be used for any publication on this topic. 

Figures 1 and 2 are well represented and explained. I have no problems with them

The 209 references are up-to-date and very well represent this field, and provide a wealth of information on this topic. I anticipate that this review will be used as a reference in any future publication on this topic.

 I highly recommend acceptance of this review, and look forward to seeing it as a publication.

This is one of the rare occasions when I see a manuscript so complete and well represented.

Response to Reviewer #1

Thank you for taking the time to review our manuscript, we are pleased that you appreciate it and found it suitable for publication in Biomedicines.

Reviewer 2 Report

Thank you for this submitted paper. However, some points should be considered/corrected.

-Title, abstract and tables should be consistent. Sometimes there is said hematological malignancies, sometimes lymphoma. Most of the text and abstract are dealing lymphomas, but in table 2 also other hematological malignancies are mentioned. The text is too long, so I would focus on lymphomas. Your main points stars after line 385, so the previous text should be only supporting to that.

-Please check the lining in table 1, some of the columns are going are in different places. 

-Table 1 should be clarified. The figure text says antibodies evaluated in lymphoma, but the table is more about approved antibodies. I would recommend to focus on approved medications in this.

-From the tables/text are missing: durvalumab, although there is a published study in HL (Casulo et al. Cancer Rep 2023; 6(1):e1662). Also for atezolizumab there is published study for HL (Geoerger B et al. Lancet oncology 2020; 21(1):134-144).

-Please clarify the text according to the table: do not repeat what is said in tables 2-3, but rather make a clearer summary in text.

-The tables 2-3 are hard to read. For example, could the left column (diagnosis), be vertical to give space to other colums? Please add the titles (studies, N, etc) to every page to ease the reading. This table is referred only once, although it has results concerning many paragraphs.

-Line 307: please change = -> -.

-Table 3 is also referred only once, altough it also has results concerning multiple paragraphs.

-The connection to the patient case is very loose. Although it might inspired the authors to write this article, I do not see any value to present it. 

-Line 437 seems to miss a reference. Line 438: Please write references in numerical order (44, 139, 147).

-LIne 504: what is the conclusion of these numbers? Rather than listing the different results in a review like this the results should be taken to a higher level.

-There are multiple lines concerning solid tumors, which are unnecessary in this context. Please remove those. 

-In the table 3, there are two "groupe" words, please change them to group. On the page 19, in LYSA trial, there is venetoclax 800 mg/j change to mg/day.

Author Response

Response to Reviewer #2:

We thank the Reviewer #2 for his relevant suggestions and we would like to address a point-by-point response to Reviewer #2 as follow (please note that the mentioned lines are from the new manuscript):

Thank you for this submitted paper. However, some points should be considered/corrected.

-Title, abstract and tables should be consistent. Sometimes there is said hematological malignancies, sometimes lymphoma. Most of the text and abstract are dealing lymphomas, but in table 2 also other hematological malignancies are mentioned. The text is too long, so I would focus on lymphomas. Your main points stars after line 385, so the previous text should be only supporting to that. ??

As suggested by Reviewer #2 in a different comment, we remove the case story, our introduction is now reduced to 407 words, this has significantly shortened our manuscript. Additionally, we considerably simplify the efficacy sections from lines 70 to 297 (see response to another comment below).

-Please check the lining in table 1, some of the columns are going are in different places. 

-Table 1 should be clarified. The figure text says antibodies evaluated in lymphoma, but the table is more about approved antibodies. I would recommend to focus on approved medications in this.

We agree with Reviewer #2 that most of PD1 antibodies are currently approved, however, PDL1 inhibitors are not yet approved in this indication. As we discussed them extensively in the manuscript, we initially found that it would be interesting to keep them in this descriptive Table 1 that also described the approval state of each antibody, however, as suggested by Reviewer #2 we remove PDL-1 inhibitors from Table 1 and replace title as follow: “currently approved PD-1 antibodies in lymphoma”.

-From the tables/text are missing: durvalumab, although there is a published study in HL (Casulo et al. Cancer Rep 2023; 6(1):e1662).

Also for atezolizumab there is published study for HL (Geoerger B et al. Lancet oncology 2020; 21(1):134-144).

As Georger et al. study mainly focus on a pediatric/young adults population we initially decide not to include it, however, we add it to the revised version as well as Casulo et al. study in order to cover all studies in this table. Furthermore we mentioned the Casulo trial in the manuscript as follow: line 259-261: “Similarly, durvalumab in combination with ibrutininb, lenalidomide +/- rituximab or bendamustine +/- rituximab only add little benefit (60).“

-Please clarify the text according to the table: do not repeat what is said in tables 2-3, but rather make a clearer summary in text.

As suggested by Reviewer #2 in several comments, we extensively reshape and simplify the efficacy sections  from lines 70 to 297 and add references to tables 2 and 3 accordingly.

-The tables 2-3 are hard to read. For example, could the left column (diagnosis), be vertical to give space to other colums? Please add the titles (studies, N, etc) to every page to ease the reading. This table is referred only once, although it has results concerning many paragraphs.

We have added the titles of the tables to every pages and change the tables to ease the reading. We also referred the tables multiple times in the manuscript accordingly Reviewer #2 comment.

-Line 307: please change = -> -.

We change = -> - accordingly at line 307.

-Table 3 is also referred only once, although it also has results concerning multiple paragraphs.

As suggested by Reviewer #2 in several comments, we simplify the text manuscript from the lines 70 to 297 and add references to tables 2 and 3 accordingly.

-The connection to the patient case is very loose. Although it might inspired the authors to write this article, I do not see any value to present it. 

We removed the patient case. Consequently, we changed the order of the paragraphs and changed the manuscript title from ” Immune checkpoint blockade and biomarkers of response in lymphoma: a case-illustrated narrative review” to “Efficacy of immune checkpoint blockade and biomarkers of response in lymphoma: a narrative review”

We also adapted the sentences referring to the clinical case in the manuscript as follow:

Lines 61-62: “Here, we present the clinical case of a patient suffering from an aggressive NHL with multiple relapses who was treated successfully with ICB. We then describe clinical efficacy of PD-1/PD-L1 ICB as monotherapy and in combination with other agents for different lymphoma subtypes.” for “Here, we present a review of the clinical efficacy of PD-1/PD-L1 ICB as monotherapy and in combination with other agents for different lymphoma subtypes.”

Lines 479-481: ”Some patients with specific NHL subtypes, (e.g. PMBCL, PTL, PCNSL) or particular molecular findings such as in the case we present (a patient with a high TMB), may derive prolonged clinical benefit from these agents.”for ”Some patients with specific NHL subtypes, (e.g. PMBCL, PTL, PCNSL) or particular molecular findings (e.g. high TMB), may derive prolonged clinical benefit from these agents.”

-Line 437 seems to miss a reference. Line 438: Please write references in numerical order (44, 139, 147).

Thank you for noticing this missing reference that we add at line 348. We also change the order of the reference as suggested at line 349 from the revised manuscript).

-Line 504: what is the conclusion of these numbers? Rather than listing the different results in a review like this the results should be taken to a higher level.

As suggested by Reviewer #2, we change the conclusion in lines 405 to 408 as follow:

“Another study, reported as an abstract only, compared TMB with pTMB in several NHL subtypes , with a respective median mut/Mb of 8.9 vs 10.0 for DLBCL, 3.5 vs 3.2 for small B cell lymphoma, 2.4 vs 2.1 for PTCL and 5.9 vs 3.2 for NK/T cell lymphoma (176).” for Another study, reported as an abstract only, compared TMB and pTMB in several NHL subtypes and concluded to a higher TMB in DLBCL than in other lymphoma subtypes . Based on these results, it seems that a higher TMB seems to correlate to PD1 ICB response as this is the case for other solid cancers (169).”

-There are multiple lines concerning solid tumors, which are unnecessary in this context. Please remove those. 

As suggested by Reviewer #2, we removed or adapted the following sentences concerning solid tumors from the new manuscript:

Lines 205-206: In solid malignancies, combination of PD-1/PD-L1 inhibitors with other ICB molecules have shown superior efficacy compared to single agent ICB (63).” has been deleted from the initial manuscript.

Lines 315-319: “In some solid malignancies, PD-L1 expression (measured by immunohistochemistry; IHC) has been associated with an improved response to PD-1/PD-L1 ICB (i.e. melanoma, non-small cell lung carcinoma (NSCLC) and urothelial carcinoma) (123–125) and worse OS irrespective of ICB exposure (126). PD-L1 expression is now routinely measured by IHC in several solid tumors (127), however, its discriminative value is likely imperfect as patients with PD-L1–negative tumors may achieve durable responses (ref). Additionally, the PD-L1 positive cut-off threshold remains highly variable depending on tumor type, with different IHC scoring systems used to determine PD-L1 expression in tumors (128).” was changed for “Different scoring systems are used to determine PD-L1 expression (measured by immunohistochemistry; IHC) in tumors, however, its discriminative value is likely imperfect as patients with PD-L1–negative tumors may achieve durable responses (124). Additionally, the PD-L1 positive cut-off threshold remains highly variable depending on tumor type, with different scoring systems used to determine PD-L1 expression in tumors (125).”

Lines 451-453: In solid tumors, e.g. NSCLC and melanoma, the level of mRNA IFN-γ expression has been demonstrated to correlate with PD-1/PD-L1 ICB response (149) has been deleted from the initial manuscript

Line 365: “Thus, viral infections (i.e. human papillomavirus (HPV), EBV) seem to play a role as potential biomarkers of response to PD-1/PD-L1 ICB in solid tumors (151,152). Higher immune cell infiltration and PD-1/PD-L1 expression levels have been described in HPV-positive solid tumors (153). Similarly, EBV-positive solid tumors and lymphomas tend to benefit from ICB therapy (154,155).” was changed for Thus, EBV-positive lymphomas tend to benefit from ICB therapy (147,148).”

Comments on the Quality of English Language

-In the table 3, there are two "groupe" words, please change them to group. On the page 19, in LYSA trial, there is venetoclax 800 mg/j change to mg/day.

We thank Reviewer#2 for notify these mistakes that have been corrected in Table 3.

According to the comment we also change the following sentence in the manuscript:

Lines 21-23: “This case-illustrated review aims to appraise the current knowledge on PD-1/PD-L1 ICB efficacy in lymphoid malignancies, when used either as monotherapy or combined with other agents, and describes potential biomarkers of response in this specific setting”. for “This review aims to appraise the current knowledge on PD-1/PD-L1 ICB efficacy in lymphoma when used either as monotherapy or combined with other agents, and describes potential biomarkers of response in this specific setting.”

Round 2

Reviewer 2 Report

Thank you for the corrections. The tables could still be more readable, but this might be an editorial question.

Acceptable.